## Research Article

barries and facilitators; implementation strategies; mental health; adoption of evidence-based practice; mental health professionals

**Palabras clave:**
Barreras y facilitadores; estrategias de implementación; salud mental; adopción de práctica basada en evidencia; profesionales de la salud mental

**Corresponding author:**
Violeta Félix-Romero;
Email: violeta.felix@psicologia.unam.mx

# How do health professionals face barriers? A quantitative approach to the adoption of evidence-based practices in mental health care

Violeta Félix-Romero[1] 🄳, Marcela Rosas-Peña[1], Diana Patricia Tzek-Salazar[1], Kalina Isela Martínez-Martínez[2] and Silvia Morales-Chainé[1]

[1]School of Psychology, Universidad Nacional Autónoma de México, México and [2]School of Psychology, Autonomous University of Aguascalientes, Mexico

## Abstract

Despite the existing evidence of effective strategies to reduce mental health risks at a reasonable cost, their adoption is still not easy for health professionals, especially in low-resource settings. Barriers and facilitators identification is then relevant for the adoption of evidence-based interventions in mental health care. The purpose of this study was to identify the relationship between barriers, facilitators and the implementation strategies to face them, related to the adoption of mhGAP Intervention Guide in primary care Mexican staff. A sample of 125 health professionals participated, after accrediting an online booster course, by answering the *Facilitators and Barriers for mhGAP Adoption Questionnaire* about the implementation of the mhGAP Intervention Guide, the implementation strategies to face those barriers and the adoption dimensions of frequency, usefulness and effectiveness of the mhGAP core components. The results revealed that *Material* was the most frequent facilitator for the implementation of mhGAP program, *Application* issues were the main barrier to its implementation and the most frequent implementation strategies reported were *Assumed the barrier* and *Tailor the intervention*, which was reported as the most effective strategy for achieving successful implementation outcomes. Barriers are discussed as important triggers for the adoption and adaptation of evidence-based practice.

## Resumen

A pesar de la evidencia existente sobre estrategias efectivas para reducir los riesgos para la salud mental a un costo razonable, su adopción aún no es fácil para los profesionales de la salud, especialmente en entornos con recursos limitados. La identificación de barreras y facilitadores es, por lo tanto, relevante para la adopción exitosa de intervenciones basadas en la evidencia para la atención de la salud mental. El objetivo de este estudio fue identificar la relación entre las barreras, los facilitadores y las estrategias de implementación para afrontarlos, en relación con la adopción de la Guía de Intervención mhGAP por parte del personal de atención primaria en México. Una muestra de 125 profesionales de la salud participó, tras completar un curso de refuerzo en línea, respondiendo el *Cuestionario sobre Facilitadores y Barreras para la Adopción de la Guía de Intervención mhGAP*. Este cuestionario abordaba los aspectos que han facilitado e impedido la implementación de la Guía de Intervención mhGAP, las estrategias de implementación para afrontar dichas barreras y las dimensiones de adopción: frecuencia, utilidad y efectividad de los componentes centrales de Guía de Intervención mhGAP. Los resultados revelaron que el *Material* fue el facilitador más frecuente para la implementación del programa mhGAP, mientras que los problemas de *Aplicación* constituyeron la principal barrera. Las estrategias de implementación más frecuentes fueron *Asumir la barrera* y *Ajustar la intervención*, esta última considerada la más eficaz para lograr resultados exitosos. Se analizan las barreras como factores clave para la adopción y adaptación de práctica basada en la evidencia.

## Impact Statements

This study reorients the implementation paradigm, positing that barriers in real clinical practice do not lead to failure but are the main engine for adoption. Our findings reveal that, especially in low-resource environments, such as the Mexican mental health system, proactive adaptation

and tailoring of interventions is not only the most common, but the most effective strategy for achieving successful implementation outcomes. Therefore, documenting these adaptations sets the direction for future research on the crucial role of health professionals' actions in reducing the mental health care gap in complex settings.

## Introduction

Mental health professionals in primary care level face the challenge of providing effective care for people who seek treatment and do it in a cost-effective way to respond to the high demand for services (World Health Organization [WHO], 2021). Despite the existing evidence of effective interventions such as screening and brief interventions to reduce mental health risks at a reasonable cost (Thoele et al., 2021; Gette et al., 2023), evidence-based treatments are provided in only 25% of the community services (McGovern et al., 2013). Moreover, it could take up to 25 years for health professionals to adopt them in their daily practice (Bauer et al., 2015).

To facilitate the delivery of evidence-based interventions in non-specialized health care settings and reduce the barriers that health staff face, the World Health Organization has developed and disseminated the Mental Health Gap Action Programme Intervention Guide (mhGAP-IG) (World Health Organization, 2016). This tool provides guidance for the assessment, management and follow-up of priority conditions, including Depression, Psychoses, Epilepsy, Child and Adolescent Mental and Behavioral Disorders, Dementia, Disorders due to Substance Use and Self/Harm Suicide. The mhGAP Programme has shown positive outcomes in low- and middle-income countries, particularly in the Latin America Region. Its implementation has led to an increase in the rates of effective detection, diagnosis and treatment of common disorders, efficient referral strategies (Miguel-Esponda et al., 2020; Sapag et al., 2021), and globally, to improve symptoms and engagement with care, patients' integration into the community, socio-emotional well-being of children and decrease in mortality by suicide (Spagnolo and Lal, 2021).

The mhGAP-IG has been widely disseminated mostly by training courses, seminars, workshops and booster sessions, resulting in the feasibility of implementing, expansion of the number of health professionals who are trained and received supervision and the improvement of their competencies (Keynejad et al., 2021). In Mexico, the National Health System has integrated the mhGAP Programme at the primary care level, through the network of psychology, medicine, nursing and social work professionals (Miguel-Esponda et al., 2020; Félix Romero et al., 2023). However, its implementation remains constrained by mental health training opportunities, a shortage specialist for supervision and contextual challenges such as non-attendance by clients because of the distance, lack of social support or stigma and lack of effective referral mechanisms, among others (Miguel-Esponda et al., 2020).

Although evidence-based interventions (EBIs), such as the mhGAP Programme, have demonstrated clinical effectiveness, the main challenge remains in their adoption, which is one of the focuses of implementation science (Curran et al., 2012). Adoption, defined as the consistent and routine implementation of EBIs in real scenarios, is influenced by multiple factors, including intervention characteristics, outer and inner settings, individual characteristics and the implementation process itself (Damschroder and Hagedorn, 2011). Although empirical research on implementation processes is still limited (Padwa and Kaplan, 2018), various implementation strategies have been identified as methods to enhance the adoption, the implementation itself and the sustainability of clinical practices (Louie et al., 2021).

With the aim of exploring the implementation strategies, recent studies have examined factors that either facilitate or impede the adoption of EBIs in mental health care in a wide variety of disorders and issues included post-traumatic stress (Finch et al., 2020), incarceration (Coffey et al., 2025), gambling (Selin et al., 2020) and alcohol and drug use (Martínez et al., 2018; Keen et al., 2021). Within implementation science, these factors are known as barriers and facilitators (Bunting et al., 2025).

Barriers and facilitators have differential implications for health professionals. On the one hand, barriers limit the effective use of interventions and, in some cases, can lead to bad practices, poor quality services and limited access to mental health resources (Barry et al., 2023). Nonetheless, health professionals make valuable efforts to face and work with them, resulting in implementation strategies that facilitate their job, such as educational outreach, specialized training, use of manuals, protocols, printed education materials, local champion leaders, supervision and feedback (Keen et al., 2021; Bunting et al., 2025). Despite their importance for improving implementation processes and outcomes (Geng et al., 2023), these actions remain underexplored (Louie et al., 2021).

Most of the studies about barriers and facilitators have employed qualitative designs, mainly focus groups and interviews, to identify the health professionals' perceptions about the implementation and adoption process (Finch et al., 2020; Han and Kim, 2024). These methods have made it possible to develop a causal model of implementation with seven main concepts identified (Leonard et al., 2020): context, innovation, relation and networks, institutions, knowledge, actors and resources. All these factors combined determine the extent to which an innovation is adopted, and specifically, the decision by health professionals to use an intervention seems to be related to their competencies, the quality of training and supervision received and the availability of manuals and protocols to guide the implementation (Leonard et al., 2020; Barry et al., 2023). Although that approach provides valuable data on the perception of professionals, it leaves out the possibility of objectively identifying the relationship between barriers and facilitators at different levels, and their effect on the implementation outcomes; consequently, quantitative studies are needed to explore this interaction.

Implementing guides such as the mhGAP-IG requires the commitment of institutions and staff to adapt the procedures to their local conditions and respond to the needs of their community. Therefore, understanding the factors that promote adoption is crucial. The present study aimed to identify the relationship between barriers, facilitators and the implementation strategies to face them, related to the adoption of mhGAP Intervention Guide in primary care Mexican staff. First, we aimed to describe the facilitators, barriers and strategies to face them reported by health professionals; second, we identified the relationship between the strategies to overcome barriers and the implementation outcomes and third, we characterized how these implementation strategies influence implementation outcomes. We hypothesized that distinct clusters would emerge based on the strategies employed. Furthermore, we expected these clusters to influence the perception of facilitators and barriers, as well as the overall level of mhGAP-IG adoption.

**Table 1.** Distribution of demographic characteristics of the sample

| Demographic characteristics | N (%) | |
|---|---|---|
| **Sex** | | |
| Female | 80 (64.0) | |
| Male | 45 (36.0) | |
| **Scholar level** | | |
| Intern | 2 (1.6) | |
| Degree | 73 (58.4) | |
| Postgraduate | 50 (40) | |
| **Institutional position** | | |
| Executive | 11 (8.8) | |
| Treatment provider | 86 (68.8) | |
| Administrative | 16 (12.8) | |
| Other | 12 (9.6) | |
| | M | SD |
| Age | 40.70 | 9.103 |
| Years of clinical experience | 9.27 | 6.536 |

## Method

### Participants

A convenience sample of 125 health professionals from a federal public institution across five Mexican states (Coahuila, Morelos, Estado de México, Guerrero and Chiapas) participated in the study. The mean age was 40.7 years ($SD = 9.1$), and 9.2 ($SD = 6.5$) years of experience in clinical settings. Table 1 shows their demographic characteristics.

Inclusion criteria were: (1) Completion of an online booster training course on the mhGAP-IG (Félix Romero et al., 2023), (2) At least 6 months of experience implementing the mhGAP-IG and (3) Being a professional in the field of psychology, medicine, nursing or social work. The participation rate was 100%, as the study was supported by institutional authorities and staff.

### Instruments

We developed a *Facilitators and Barriers for mhGAP Adoption Questionnaire*, considering the model proposed by Leonard et al. (2020), the Consolidated Framework for Implementation Research (Damschroder et al., 2009; Damschroder and Hagedorn, 2011), the Expert Recommendations for Implementing Change (ERIC) project (Powell et al., 2015) and previous research developed in Mexico about barriers for the implementation of brief interventions (Martínez et al., 2018).

The instrument is composed of five sections, and, for each section, a confirmatory factor analysis was conducted to confirm the models:

### Facilitators

This section assessed aspects that promote the use of the mhGAP strategy (Padwa and Kaplan, 2018). Participants were asked: "*Indicate from 1 to 5 (where 1 means not at all and 5 means completely) to what extent the following aspects have facilitated the implementation of the mhGAP-IG.*" The analysis yielded three factors ($\alpha = 0.94$; $X^2(109) = 231.34$; $p = <.001$; CFI = 0.94, RMSEA = 0.081, confidence intervals from 0.067 to 0.096; SRMR = 0.058): *Material Facilitators*

(seven items about the availability of manuals, brochures, records and infrastructure resources), *Training Facilitators* (four items about quality and quantity of training, supervision and interaction with other professionals and institutions) and *Application Facilitators* (six items about steps and components of the intervention and intervention format such as online or face to face delivery).

### Barriers

Explores the aspects that participants consider have made the implementation of the mhGAP-IG difficult (Leonard et al., 2020). In this section, participants were asked: "*Indicate from 1 to 5 (where 1 means not at all and 5 means completely) to what extent the following aspects represent a barrier to the implementation of the mhGAP-IG.*" This section yielded five factors ($\alpha = 0.89$; $X^2(607) = 905.84$; $p < .01$; CFI = 0.93, RMSEA = 0.054, confidence intervals from 0.046 to 0.061; SRMR = 0.064): *Application Barriers* (10 items about how the intervention is implemented, duration, number of steps, sequence), *Material Barriers* (five items about availability of manuals, brochures, records and infrastructure resources), *Training Barriers* (eight items about quality and quantity of training, supervision and interaction with other professionals and institutions), *Client Barriers* (eight items about characteristics of clients) and *Treatment Provider Barriers* (six items about characteristics of health professionals).

### Adoption dimensions

The next three sections are about the Adoption Dimensions, exploring the extent to which the intervention is being used, including frequency, usefulness and effectiveness of the mHGAP-IG core components (General principles, Essentials of mental health practice, Treatment planning, Psychosocial interventions, Pharmacological interventions, Referral and Follow-up).

*Implementation Frequency* ($\alpha = 0.94$; $X^2(91) = 261.08$ $p < .001$; CFI = 0.92, RMSEA = 0.105, confidence intervals from 0.090 to 0.120; SRMR = 0.058). In this section, participants were asked in 17 items about the frequency of use of the main components of the mhGAP-IG: "*The core components of the mhGAP strategy are listed below. Please indicate how frequently you implement each of them.*"

*Implementation Usefulness* ($\alpha = 0.96$; $X^2(90) = 299.46$; $p < .001$; CFI = 0.93, RMSEA = 0.117, confidence intervals from 0.103 to 0.132; SRMR = 0.039). In this section, participants were asked in 17 items about the usefulness of the main components of the mhGAP-IG: "*The core components of the mhGAP strategy are listed below. Please indicate how useful their implementation has been for you.*"

*Implementation Effectiveness* ($\alpha = 0.96$; $X^2(93) = 291.83$; $p < .001$; CFI = 0.94, RMSEA = 0.112, confidence intervals from 0.098 to 0.127; SRMR = 0.039). In this section, participants were asked 17 items about the effectiveness of the main components of the mhGAP-IG: "*The core components of the mhGAP strategy are listed below. Please indicate how effective this component has been with the people you have worked with.*"

Finally, participants identified actions used to address barriers and enhance mhGAP-IG adoption, grouped in five areas: *Application Barriers* (10 items), *Material Barriers* (five items), *Training Barriers* (eight items), *Client Barriers* (eight items) and *Treatment Provider Barriers* (six items). The answer options included the implementation strategies (Powell et al., 2015): *Tailor the intervention* (make adjustments based on specific needs and conditions), *Assume the barrier* (promote adaptability and facilitation to team problem solving and meet local needs), *Impede the implementation* (the extent that the identified barrier makes implementation impossible) or *Not considering a barrier* (the issue identified is not considered a barrier).

## Procedure

The study was conducted within the Mexican Health System, which operates a national network of public primary care clinics for mental health and addictions. Since 2023, these clinics have provided prevention and treatment services in every state of the country, which presents a challenge due to the diversity of communities and socioeconomic conditions that prevail in Mexico. Since their creation, the task force, comprised of psychologists, medical doctors, nurses and social workers, has been training by courses and seminars and is responsible for implementing interventions based on mhGAP-IG.

In this context, we invited participants through institutional channels with administrative support. Participants voluntarily agreed to participate, provided informed consent and were offered a participation certificate. Consent forms emphasized confidentiality, research use of data, benefits and minimum risks about the activities involved in their participation.

Six months after participants successfully finished the mhGAP online booster, consisting of 40 hours of review on the Moodle® platform covering the fundamentals and procedures for essential care and practice, including the assessment, management and follow-up of priority mental, neurological and substance use disorders, they were asked *via* email to complete the *Facilitators and Barriers for mhGAP Adoption Questionnaire* using Google Forms®. The questionnaire required approximately 30 min to complete, and participants were given 1 week to submit their responses (Supplementary Appendix).

## Data analysis

We followed a transversal design with quantitative analyses, organizing the data analysis into three parts: (1) descriptive analysis about the frequency of facilitators, barriers, implementation strategies and adoption dimensions; (2) correlational analysis about the relationship between the implementation strategies and adoption dimensions and (3) to test the hypothesis, a cluster analysis was performed to identify distinct patterns among participants based on the implementation strategies used to address barriers related to the adoption of the mhGAP-IG. This analytical approach enables the identification of naturally occurring subgroups derived from shared characteristics, without imposing any *a priori* classification. The *k-means* algorithm was applied as the clustering method, using squared Euclidean distances as the measure of dissimilarity. The optimal number of clusters was determined through the inverse scree technique. The proportion of use of the four implementation strategies (*Tailor the intervention, Assume the barrier, Impede the implementation and Not considering it a barrier*) across the five barrier areas was entered into the analysis. Subsequently, a multivariate analysis was conducted among the groups derived from the cluster solution, according to the coping strategy employed. An analysis of variance (ANOVA) was then performed to examine the effect of the coping group on the perception of barriers, facilitators and program implementation.

## Results

### Descriptive analysis of the distribution of facilitators, barriers, strategies to face barriers and implementation dimensions

As shown in Table 2, the descriptive analysis indicated that *Material* was the most frequent facilitator for implementing mhGAP-IG ($\mu = 82.0$, $SD = 14.1$). On the contrary, *Application* issues turned out

**Table 2.** Distribution of the average percentage of participants' perception about the extent of facilitators, barriers, adoption dimensions and implementation strategies to face barriers

| Section | Factor | Total M | SD |
|---|---|---|---|
| Facilitators | Material | 82.0 | 14.1 |
| | Training | 68.9 | 18.5 |
| | Application | 68.8 | 17.1 |
| Barriers | Application | 57.0 | 16.4 |
| | Material | 54.0 | 26.7 |
| | Training | 49.2 | 13.6 |
| | Client | 48.3 | 15.1 |
| | Treatment provider | 45.3 | 16.4 |
| Adoption dimension: frequency | | 80.1 | 13.9 |
| Adoption dimension: usefulness | | 89.5 | 11.2 |
| Adoption dimension: effectiveness | | 88.1 | 11.7 |
| Implementation strategy to face barriers | | | |
| Tailor the intervention | | 30.15 | 1.27 |
| Assume the barrier | | 42.35 | 1.09 |
| Impede the implementation | | 13.52 | 0.84 |
| Not a barrier | | 14.04 | 0.80 |

to be the main barrier to its implementation ($\mu = 57.0$, $SD = 16.4$). Regarding the frequency, usefulness and effectiveness dimensions, participants reported a high level of adoption, between 80.1% and 89.5%, in the three dimensions.

Once the barriers were identified, participants reported the strategies they used to address them. The main strategy reported was *Assumed the barrier* ($\mu = 42.35$, $SD = 1.09$), followed by *Tailor the intervention* ($\mu = 30.15$, $SD = 1.27$), *Not a barrier* ($\mu = 14.04$, $SD = 0.80$) and *Impede the implementation* ($\mu = 13.52$, $SD = 0.84$).

### Correlational analysis of the relationship between mhGAP-IG core components and their facilitators, barriers and implementation strategies to address barriers

Table 3 presents the correlational analysis between Facilitators, Barries, Adoption dimensions and Implementation Strategies to address barriers. The results revealed low but significant relations, suggesting that *Material* acted as a key facilitator of mhGAP-IG implementation, correlating positively with *Tailor the intervention* ($r = 0.23$) and negatively with *Assume the barrier* ($r = -0.18$).

High levels of *Treatment providers* barriers were negatively related to *Usefulness* ($r = -0.20$) and *Effectiveness* ($r = -0.19$) of mhGAP-IG implementation. In addition, barriers related to *Material* ($r = 0.24$) and *Treatment providers* ($r = 0.18$) were linked to a greater tendency to *Assume that barriers* rather than *Tailor the intervention* ($r = -0.29$). Conversely, more barriers related to *Clients* ($r = -0.34$) were associated with fewer *Tailoring* actions by health professionals.

Notably, the *Assume* strategy to face barriers is negatively related to the adoption dimensions of *Frequency* ($r = -0.18$), *Usefulness* ($r = -0.23$) and *Effectiveness* ($r = -0.24$). In contrast, *Tailor the intervention* correlated positively with *Usefulness* ($r = 0.30$) and *Effectiveness* ($r = 0.35$).

**Table 3.** Correlational analysis between the use of core components of the mhGAP-IG and their facilitators, barriers and implementation strategies to face barriers

| Variable | 1 | 2 | 3 | 4 | 5 | 6 | 7 | 8 | 9 | 10 | 11 | 12 | 13 | 14 | 15 | 16 |
|---|---|---|---|---|---|---|---|---|---|---|---|---|---|---|---|---|
| 1 | — | | | | | | | | | | | | | | | |
| 2 | 0.672*** | — | | | | | | | | | | | | | | |
| 3 | 0.7*** | 0.794*** | — | | | | | | | | | | | | | |
| 4 | −0.027 | −0.011 | −0.027 | — | | | | | | | | | | | | |
| 5 | −0.147 | −0.027 | −0.061 | 0.512*** | — | | | | | | | | | | | |
| 6 | −0.031 | 0.05 | 0.058 | 0.08 | 0.039 | — | | | | | | | | | | |
| 7 | −0.026 | −0.003 | 0.004 | 0.063 | 0.079 | 0.338*** | — | | | | | | | | | |
| 8 | −0.02 | 0.008 | 0.024 | 0.057 | 0.214* | 0.16 | 0.477*** | — | | | | | | | | |
| 9 | −0.11 | −0.052 | −0.116 | −0.078 | −0.06 | 0.08 | −0.134 | −0.152 | — | | | | | | | |
| 10 | 0.088 | 0.064 | −0.057 | −0.042 | −0.059 | 0.077 | −0.064 | −0.207* | 0.545*** | — | | | | | | |
| 11 | 0.059 | 0.04 | −0.044 | 0.056 | −0.03 | 0.121 | −0.073 | −0.194* | 0.554*** | 0.844*** | — | | | | | |
| 12 | 0.234** | 0.074 | 0.106 | 0.04 | −0.171 | −0.142 | −0.344*** | −0.299*** | 0.127 | 0.305*** | 0.356*** | — | | | | |
| 13 | −0.181* | −0.036 | −0.05 | 0.121 | 0.234** | 0.068 | 0.17 | 0.186* | −0.181* | −0.23* | −0.242* | −0.704*** | — | | | |
| 14 | −0.094 | −0.11 | −0.133 | −0.122 | −0.105 | 0.046 | 0.151 | 0.055 | 0.172 | −0.046 | −0.024 | −0.315*** | −0.237** | — | | |
| 15 | $4.522 \times 10^{-4}$ | 0.05 | 0.043 | −0.154 | 0.014 | 0.091 | 0.173 | 0.164 | −0.105 | −0.105 | −0.200* | −0.248** | −0.232** | −0.095 | — | |
| 16 | 0.104 | 0.053 | −0.043 | −0.074 | −0.199* | 0.025 | 0.116 | −0.004 | 0.133 | 0.225* | 0.218* | 0.085 | −0.018 | #### | −0.071 | — |

*Note*: 1. Material facilitators, 2. Training facilitators, 3. Application facilitators, 4. Application facilitators, 5. Material barriers, 6. Training barriers, 7. Client barriers, 8. Treatment provider barriers, 9. Adoption dimension: Frequency, 10. Adoption dimension: Usefulness, 11. Adoption dimension: Effectiveness, 12. Tailor the intervention, 13. Assume the barrier, 14. Impede implementation, 15. Not a barrier, 16. Years of experience.
*Significant <.05; **Significant <.01; ***Significant <.001

Finally, years of experience in the mental health field were positively associated with perceived *Usefulnees* ($r = 0.22$) and *Effectiveness* ($r = 0.21$).

## Analysis of variance for different implementation strategies used to face barriers

Consistent with the correlational analysis, a relationship emerged between the implementation strategies used to face barriers, the barriers and facilitators perceived and the adoption dimensions. To explore the effect of these strategies, a cluster analysis was conducted to group participants based on the implementation strategies they reported (Table 4). The K-mean cluster analysis identified five distinct cluster centers demonstrated distinct groupings: Cluster 1 was characterized by high levels of *Tailor the intervention* strategy ($\mu = 65.9$, $SD = 13.1$), Cluster 2 combined high levels of *Tailor the intervention* ($\mu = 34.8$, $SD = 9.9$) and *Assume the barrier* ($\mu = 49.3$, $SD = 7.9$) strategies, Cluster 3 has high levels of *Assume the barrier* strategy ($\mu = 76.3$, $SD = 10.0$), Cluster 4 was represented by high levels of *Not a barrier* report ($\mu = 46.3$, $SD = 16.9$) and finally, Cluster 5 was characterized by high levels of *Impede the implementation* ($\mu = 36.3$, $SD = 9.2$) and *Assume the barrier* strategies ($\mu = 36.7$, $SD = 12.9$).

ANOVA results (Table 5) revealed significant differences between clusters in Client Barriers ($F(4, 124) = 2.45$, $p = 0.05$), Effectiveness of Adoption dimension ($F(4, 124) = 3.08$, $p = 0.01$)

**Table 4.** K-means clustering

| Clusters[a] | N | R² | AIC | BIC | Silhouette |
|---|---|---|---|---|---|
| 5 | 124 | 0.712 | 181.8 | 238.2 | 0.32 |
| Cluster[b] | 1 | 2 | 3 | 4 | 5 |
| Size | 31 | 24 | 26 | 8 | 35 |
| Explained proportion within-cluster heterogeneity | 0.243 | 0.12 | 0.263 | 0.154 | 0.219 |
| Within sum of squares | 34.517 | 16.987 | 37.293 | 21.891 | 31.106 |
| Center AJ | 1.352 | −0.965 | −0.6 | −0.583 | 0.044 |
| Center AS | −0.997 | 1.544 | −0.311 | −0.798 | 0.238 |
| Center IM | −0.337 | −0.635 | 1.582 | −0.379 | −0.354 |
| Center NB | −0.35 | −0.282 | −0.075 | 2.984 | −0.123 |

*Note:*
[a]The model is optimized with respect to the *BIC* value.
[b]The Between Sum of Squares of the 5 cluster model is 350.2; The Total Sum of Squares of the 5 cluster model is 492.

**Table 5.** Analysis of variance of facilitators, barriers, adoption dimensions and Implementation strategies by clusters

| Area | | Total | | Group 1 (n = 31) | | Group 2 (n = 35) | | Group 3 (n = 24) | | Group 4 (n = 8) | | Group 5 (n = 26) | | F (4,124) | p | η² |
|---|---|---|---|---|---|---|---|---|---|---|---|---|---|---|---|---|
| | | M | SD | M | SD | M | SD | M | SD | M | SD | M | SD | | | |
| Facilitators | Material | 82.0 | 14.1 | 84.5 | 12.0 | 83.2 | 13.2 | 79.0 | 12.5 | 81.1 | 21.2 | 79.6 | 16.3 | 0.77 | 0.546 | 0.03 |
| | Training | 68.9 | 18.5 | 69.5 | 19.2 | 68.3 | 18.6 | 69.6 | 14.1 | 70.6 | 19.5 | 66.7 | 21.4 | 0.13 | 0.973 | 0.00 |
| | Application | 68.8 | 17.1 | 71.2 | 15.9 | 65.9 | 17.5 | 71.0 | 13.6 | 70.8 | 18.6 | 66.2 | 19.7 | 0.69 | 0.603 | 0.02 |
| Barries | Application | 57.0 | 16.4 | 59.1 | 21.1 | 57.9 | 13.4 | 60.1 | 11.5 | 43.3 | 15.1 | 53.8 | 16.3 | 2.10 | 0.086 | 0.07 |
| | Material | 54.0 | 26.7 | 53.2 | 30.8 | 51.4 | 26.9 | 61.5 | 17.9 | 47.5 | 28.4 | 51.7 | 26.8 | 0.73 | 0.574 | 0.02 |
| | Traning | 49.2 | 13.6 | 46.7 | 13.2 | 49.8 | 12.3 | 50.5 | 12.8 | 52.2 | 11.0 | 49.4 | 17.0 | 0.42 | 0.794 | 0.01 |
| | Client | 48.3 | 15.1 | 41.5 | 17.4 | 50.2 | 14.4 | 48.0 | 14.4 | 52.2 | 14.3 | 52.5 | 12.2 | 2.45 | 0.050 | 0.08 |
| | Treatment provider | 45.3 | 16.4 | 38.1 | 16.3 | 47.5 | 14.3 | 46.1 | 16.7 | 48.3 | 13.8 | 48.6 | 18.3 | 2.08 | 0.088 | 0.07 |
| Adoption dimensión: frequency | | 80.1 | 13.9 | 83.7 | 12.1 | 76.3 | 14.6 | 79.0 | 19.9 | 77.0 | 8.1 | 82.6 | 8.2 | 1.54 | 0.196 | 0.05 |
| Adoption dimensión: usefulness | | 89.5 | 11.2 | 93.6 | 8.2 | 88.8 | 9.0 | 87.5 | 16.9 | 85.3 | 7.2 | 88.6 | 10.7 | 1.61 | 0.176 | 0.05 |
| Adoption dimensión: effectiveness | | 88.1 | 11.7 | 93.9 | 8.5 | 87.1 | 8.7 | 86.1 | 16.8 | 81.9 | 3.6 | 86.4 | 12.7 | 3.08 | 0.019 | 0.09 |
| Implementation strategy to face barriers | Tailor the intervention | 30.15 | 1.27 | 65.9 | 13.1 | 34.8 | 9.9 | 10.8 | 9.9 | 19.9 | 18.5 | 19.5 | 13.9 | 85.24 | 0.000 | 0.74 |
| | Assume the barrier | 42.35 | 1.09 | 20.9 | 10.6 | 49.3 | 7.9 | 79.3 | 10.0 | 25.5 | 14.5 | 36.7 | 12.9 | 114.36 | 0.000 | 0.79 |
| | Impede implementation | 13.52 | 0.84 | 9.1 | 9.5 | 8.9 | 7.3 | 4.9 | 5.9 | 8.5 | 8.1 | 36.3 | 9.2 | 63.63 | 0.000 | 0.68 |
| | Not barrier | 14.04 | 0.80 | 4.2 | 5.5 | 7.1 | 7.6 | 5.0 | 5.7 | 46.3 | 16.9 | 7.7 | 8.1 | 50.86 | 0.000 | 0.63 |

and the strategies *Tailor the intervention* ($F(4, 124) = 85.24$, $p < 0.01$), *Assume the barrier* ($F(4, 124) = 114.36$, $p < 0.01$), *Impede the implementation* ($F(4, 124) = 63.63$, $p = 0.01$) and *Not a barrier* ($F(4, 124) = 50.86$, $p = 0.01$).

*Post hoc* Games–Howell analysis showed that the *Tailor intervention* was reported more frequently by Cluster 1 ($\mu = 65.9$, $SD = 13.1$) than by the rest of the clusters, and was reported more frequently by Cluster 2 than by Clusters 3 and 4. *Assumed the barrier* strategy predominated in Cluster 2 ($\mu = 79.3$, $SD = 10.0$) over Cluster 4 and 5 ($\mu = 36.7$, $SD = 12.9$), and was significantly less used by Cluster 1 than Clusters 2, 3 and 5. Cluster 2 used the *Assume the barrier* strategy than Cluster 1, but more than Clusters 4 and 5. *Impede the implementation* was significantly more used by Cluster 5 ($\mu = 36.3$, $SD = 9.2$) than the other Clusters, and *Not a barrier* was used more in Cluster 4 ($\mu = 46.3$, $SD = 16.9$) than the rest of the Clusters.

Notably, *Post hoc* Games–Howell analysis showed that participants in Cluster 1, in which *Tailor the intervention* was the main strategy to face barriers, perceive less barriers related to clients ($\mu = 41.5$, $SD = 17.4$) than participants in Cluster 5, who reported that barriers impede the implementation.

Importantly, participants in the Cluster 1 also reported significative more effectiveness in the implementation ($\mu = 93.9$, $SD = 8.5$) that could be related to the adjustments they make than participants in the Cluster 2 who mainly assume the barrier ($\mu = 87.1$, $SD = 8.7$) and Cluster 4 where there are not barriers reported ($\mu = 81.9$, $SD = 3.6$).

## Discussion

Implementation barriers have traditionally been considered as negative aspects that must be prevented or eliminated to successfully adopt a health care intervention. However, when analysis extends beyond merely identifying barriers and instead examines how professionals actively address them, it is possible to understand the adoption and adaptation processes that underline effective implementation despite the difficulties that arise in real-life scenarios (Sims-Rhodes et al., 2024; Wijekoon Mudiyanselage et al., 2024). Focusing on the strategies that health professionals apply to overcome these barriers and their effect on the implementation of the mhGAP-IG is one of the contributions of the present work.

Consistent with the literature, especially from low- and middle-income countries (Martínez et al., 2018; Wijekoon Mudiyanselage et al., 2024; Coffey et al., 2025), this study identified two types of barriers related to the application of intervention components and those concerning the availability of material resources. Barriers in the application may reflect a lack of mastery of the core components responsible for the intervention's effectiveness and, therefore, may jeopardize its fidelity (Rabin et al., 2008). In contrast, insufficient material resources are a common problem in health systems in Latin America and are related to the quality and accessibility of mental health care (Keynejad et al., 2018; Sapag et al., 2021). These predominantly contextual barriers must be recognized and addressed comprehensively by health systems to bring Evidence-Based Interventions closer to the population.

Although participants report perceiving important barriers in different areas, the most relevant result is what they do about them. Notably, the last thing they do is not implement the mhGAP-IG; on the contrary, identifying barriers strongly leads to knowing that there is a problem, and, in general, assumes the challenge and makes any necessary adjustments to achieve high levels of effectiveness and usefulness (Minian et al., 2019). Our findings suggest that when health professionals perceive deficiencies in material resources and in their own skills and experience, it is negatively related to frequency, usefulness and effectiveness of the mhGAP-IG. On the other hand, when health professionals have sufficient material for their activities, they are better able and more confident to tailor the mhGAP-IG. These findings highlight the importance of adaptations made by health professionals in practice; therefore, future research should focus on identifying how an intervention can be tailored to meet local needs to promote adaptability (Proctor et al., 2013).

According to our results, tailoring is the most frequent and effective implementation strategy to address barriers. Adapting protocols is a natural part of the process of adopting an intervention and implies that health professionals must make decisions on a day-to-day, and case-by-case basis to adapt to the needs of their clients, institutions and communities (Renet et al., 2024; Toal-Sullivan et al., 2024). Research into how professionals make these decisions is needed, as it will allow us to identify exactly what they base these decisions on, and which elements of the intervention must be maintained to preserve fidelity and effectiveness (Powell et al., 2015). This has implications for training and supervision practices, as training is one of the most frequently reported implementation strategies (Louie et al., 2021), and in this study, we found it an important facilitator. Training methods must promote competencies to balance between fidelity and adaptation.

Some studies have reported that fidelity is the best way to effectively enhance clinical outcomes (Louie et al., 2021). However, contrary to that, we found that tailoring the intervention is related to more effective implementation and with fewer barriers perceived, particularly those barriers related to clients' characteristics. This adds to what is known about adaptation, defined as the extent to which an EBI is modified by clinicians during the implementation to fit the local conditions (Rabin et al., 2008; Powell et al., 2015), and the need to be understood in the context of a particular setting. Our data suggest that Mexican health professionals are practicing with a lack of resources, poor relevant training and difficulties in adhering to protocols, due to the diverse and complexed conditions of their communities. Under these conditions, tailoring the intervention seems to be the best and most effective way to achieve the implementation of mhGAP-IG. This necessity to adapt interventions creates opportunities to innovate and think about methods that account for how and to what extent these changes work, and therefore, it allows guiding the decisions of professionals and their institutions.

A distinctive contribution of this work is the quantitative approach to the analysis of barriers and facilitators, since it has traditionally been addressed through qualitative strategies (Kools et al., 2024; Akhtar et al., 2025). However, the limitation of self-reporting persists (Sims-Rhodes et al., 2024). Although self-report data are valid and informative, it is important to consider that the measures related to implementation must include direct observation, objective indicators and permanent products of the documentation of the cases attended to. Systematically documenting adaptations to understand their effects is central to implementation science (Geng et al., 2023) and can yield reciprocal benefits: it both strengthens the empirical evidence base and validates health professionals' efforts by integrating their real-world experiences into scientific feedback loops.

In conclusion, to understand the adoption process of mental health care protocols, such as the mhGAP-IG, it is necessary to contextualize barriers and implementation strategies in the daily

work of health professionals who have managed to implement EBIs despite numerous contextual challenges. Barriers, then, rather than making work impossible, become triggers for adoption and adaptation actions that must be considered and evaluated in future research as important contributions from the treatment providers themselves to clearly understand the translation process that evidence-based practice goes through. In this work, health professionals and their context and actions are placed at the center of the analysis, contributing to a more comprehensive view of the scope of adoption.

**Open peer review.** To view the open peer review materials for this article, please visit http://doi.org/10.1017/gmh.2026.10159.

**Supplementary material.** The supplementary material for this article can be found at http://doi.org/10.1017/gmh.2026.10159.

**Data availability statement.** The data that support the findings of this study are available from the corresponding author upon request.

**Author contribution.** Conceptualization: V.F.-R.; Funding Acquisition and Supervision: S.M-C.; Project Administration and Investigation: D.P.T.S.; Writing—Original Draft and Methodology: V.F.-R.; Writing—Review and Editing: K.I.M-M.; Writing—Review, Editing and Data Curation: M.R-P. All authors have read and agreed to the published version of the manuscript.

**Financial support.** This research was supported by the Gonzalo Río Arronte Foundation [SA-452] and by PAPIIT UNAM [TA300124].

**Competing interests.** The authors have no financial or non-financial interest to disclose. The funders had no role in the design of the study; in the collection, analyses or interpretation of data; in the writing of the manuscript; or in the decision to publish the results.

**Ethical standard.** Approval was obtained from the Faculty of Psychology Ethics Committee of the National Autonomous University of Mexico by number FPSI/422/CEI/277/2023. The procedures used in this study were conducted with ethical standards for human research.

**Informed consent statement.** Informed consent was obtained from all participants involved in the study.

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
