## [Reviewer Report]

Thanks for the opportunity to review. This study addresses the important issue of EBP adoption in mental health. Specifically, the lack of research findings that can be applied to improve implementation. Authors note that a good majority of implementation science is descriptive in nature and does not evaluate relationships or causation. This gap leads to the purpose of the study focused on examining relationships between barriers, facilitators, and strategies related to EBP adoption. Overall, the manuscript could be improved by adding clarity to the EBP change, terminology, and procedures. Additionally, the discussion needs quite a bit of work to compare/contrast with existing literature and clarify what others should do based on study findings. Point-by-point comments are below for each section.

Impact Statement

-As currently written, there are 3 separate statements. The author guidelines specify a short summary of beneficial use of the research presented. Recommend editing to one concise summary that captures the main statements cohesively.

Background

-Page 2, end of the last paragraph – unsure what is meant by “objectively knowing the parameters . . . “ This statement needs more clarity. It seems that authors may be trying to articulate the lack of analysis evaluating the relationship between barriers, facilitators, implementation strategies, and implementation outcomes. Intervention effectiveness seems out of place considering the intervention is described as an already established evidence-based practice.

-Page 3, first and second full paragraphs – these need to have a tighter connection to the specific problem/gap that this study addresses – These paragraphs should help logically lead the reader to the need for the current study

-More information is needed to substantiate the mhGAP-IG as an evidence-based intervention with supporting positive outcomes. Clarity is also needed for what procedures are needed that would determine adoption and fidelity of the intervention/protocol – How do we know if clinicians are deciding to use the intervention or using as intended? What needs to happen?

-Overall, the background needs to be tightened up to connect the broad problem (adoption of EBP in mental health), specific problem (implementation science can help, but unsure how to apply findings), gap (current research descriptive, little is known about relationships or causation), leading to study purpose.

-Study purpose, It would be more clear to add specific aims that align with your analysis plan and results

Methods

-Participants, include your inclusion/exclusion criteria here and clarify what disciplines were included. Did you have a target sample size based on study aims? Was 125 the participants or the recruitment sample? What was your participation rate?

-Instruments, adoption questionnaire – What resources informed development of your instrument? How many items? It might also be helpful to include brief descriptions for the included variables. For example, “material” is used throughout and it is unclear what that term means. Other terms that may be confusing are assume the barrier and impede the implementation.

-Unclear what is meant by “implementation dimensions”

-Where did the included strategy names come from? There are commonly accepted strategy names and definitions available – See A refined compilation of implementation strategies: results from the Expert Recommendations for Implementing Change (ERIC) project | Implementation Science | Full Text

-It would also be helpful to know how “adoption” was defined for the purpose of the study; Was there a measure for adoption of the intervention in practice?

Results

-Overall, results are difficult to interpret without having clear definitions/descriptions for the terms mentioned above.

Discussion

-A compare/contrast with existing literature is lacking; It is unclear how this work is specifically contributing to the body of knowledge

-What are the implications of this work for clinicians, researchers, etc.?

---

## [Reviewer Report]

The present article reports an association study between barriers and facilitators for the implementation of a mental health screening strategy and the strategies to address them in a population of health professionals in Mexico.

I consider that the manuscript addresses a relevant topic for the prevention and detection of mental health problems; however, several sections require clarification. Below are my comments:

It is necessary to clarify in more detail what is being implemented, that is, the characteristics of the training, how it was carried out, whether participation was voluntary or mandatory, and any other relevant information to better understand the type of implementation being referred to.

It is also necessary to provide much more detail about the general characteristics of the health system or the facilities in which the population carries out their professional activities. Additionally, please explain in greater detail whether there is relevant information about the implementation of mhGAP or other detection strategies in Mexico. This would help to better contextualize the implementation problem presented.

The procedure for developing the adoption questionnaire needs to be described in greater detail. Although confirmatory factor analysis data are presented, the statistical information is not sufficient to understand whether the content is relevant for measuring the intended constructs. If possible, the inclusion of the instrument as supplementary material is recommended.

The objective of the cluster analysis and the analysis of variance is not clear. It is necessary to specify in more detail the research objectives or hypotheses related to these procedures. Furthermore, please clarify the rationale for their selection and provide the specifications for their application in the data analysis section.

Finally, the conclusion only refers to the overall importance of understanding barriers; however, it does not address the specific findings of the study or explain how they point toward potential new lines of research.

---

## [Reviewer Report]

Thank you for the opportunity to review this paper a second time. Overall, authors did an excellent job resolving reviewer comments. In particular, evidence added supporting the intervention, details about the data collection instruments, and compare/contrast with the literature were much improved. I have a few minor recommendations for consideration.

Background

-Paragraph 1, second sentence – replace “strategies” with “interventions” – it needs to be clear to the reader when you are discussing interventions vs. implementation (e.g., barriers, facilitators, implementation strategies); Similarly, paragraph 3 on page 5 of the PDF, remove “tailored interventions” to minimize confusion

-Implementation science should not be capitalized; Check capitalization throughout the manuscript

-I would recommend moving discussion of the intervention of focus, mhGAP, up earlier in the background – the section currently goes back and forth between intervention and implementation, which may be difficult for the reader to follow. It would improve flow to ground the reader in the intervention of interest and then dive into the challenge of implementation in more detail.

-Page 6, paragraph 2 – the systematic review description is the only thing discussed and it is unclear how the findings relate to the current study. Consider integrating with the paragraph previous to clarify what is known/not known.

-Page 7, second full sentence from the top, replace “proven” with “has led to”

Conclusion

-Recommend adding a concluding statement (1-2 sentences) about how this study met the need described in the first sentence

---

## [Reviewer Report]

I appreciate your thoughtful review of my comments and revisions align nicely with the shared recommendations. The rationale for the work and procedures is now clear and concise. Best wishes on the next steps with this manuscript.

I did note a couple of minor opportunities for improvement with this final review:

-Page 3 of PDF, Intro, second paragraph – Review for appropriate capitalization

-Page 5 of PDF, stand alone paragraph, beginning with “Although that approach provides . . . “ Recommend splitting into 2 sentences and integrating with paragraph above or below.

---

## [Editor Report]

The manuscript is deemed acceptable subject to some minor editorial comments by the reviewer and a thorough editing by the co-authors.